# Rethinking Knowledge Transfer in Learning Using Privileged Information

**Danil Provodin**                                          *d.provodin@tue.nl*
*Eindhoven University of Technology*

**Bram van den Akker**                        *bram.vandenakker@booking.com*
*Booking.com*

**Christina Katsimerou**                    *christina.katsimerou@booking.com*
*Booking.com*

**Maurits Kaptein**                                        *m.c.kaptein@tue.nl*
*Eindhoven University of Technology*

**Mykola Pechenizkiy**                                *m.pechenizkiy@tue.nl*
*Eindhoven University of Technology*

**Reviewed on OpenReview:** *https://openreview.net/forum?id=dg1tqNIWg3*

## Abstract

In supervised machine learning, privileged information (PI) is information that is unavailable at inference, but is accessible during training time. Research on learning using privileged information (LUPI) aims to transfer the knowledge captured in PI onto a model that can perform inference without PI. It seems that this extra bit of information ought to make the resulting model better. However, finding conclusive theoretical or empirical evidence that supports the ability to transfer knowledge using PI has been challenging. In this paper, we critically examine the assumptions underlying existing theoretical analyses and argue that there is little theoretical justification for when LUPI should work. We analyze two main LUPI methods – generalized distillation and marginalization with weight sharing – and reveal that apparent improvements in empirical risk may not directly result from PI. Instead, these improvements often stem from dataset anomalies or modifications in model design misguidedly attributed to PI. Our experiments for a wide variety of application domains further demonstrate that state-of-the-art LUPI approaches fail to effectively transfer knowledge from PI. Thus, we advocate for practitioners to exercise caution when working with PI to avoid unintended inductive biases.

## 1 Introduction

In supervised machine learning (ML), we aim to learn the fit between some features $x \in \mathcal{X}$ and target $y \in \mathcal{Y}$. The information going into $x$ can only be used if it is accessible at the time of inference. However, there may exist features $z \in \mathcal{Z}$ that are only available during training due to engineering complexities or because this information only materializes post-inference. These features $z$ can present themselves in many forms, including uncompressed features (e.g., images), third-party expert annotations, non-target post-inference signals (e.g., clicks or dwell time), and metadata about the annotator/label provider. Our work is motivated by a common e-commerce application of optimizing a north-star metric, such as product conversion. [1] In

---

[1]The term "north-star metric" refers to a single key metric that serves as the primary indicator of an organization's long-term success, guiding strategy and decision-making. In the context of e-commerce, "product conversion" often serves as such a metric. Product conversion measures the rate at which users perform a desired action, such as purchasing a product, and is a critical

this context, user interactions that occur after prediction, such as clicks, can be strong indicators of the user's intent to purchase, and evaluating the probability of conversion conditioning on a click becomes more straightforward. However, clicks exist as features only in the offline data and not during inference.

For this reason, Vapnik & Vashist (2009) introduced the paradigm of learning using privileged information (LUPI). Since its introduction, LUPI has sparked significant interest within the research community across various domains, including speech recognition (Markov & Matsui, 2016), computer vision (Garcia et al., 2020; Lee et al., 2020), semi-supervised learning (Gong et al., 2018; Yang et al., 2022b), noisy-labels (Collier et al., 2022; Ortiz-Jimenez et al., 2023), and others (Li et al., 2020; Xu et al., 2020). Given this widespread interest, it is crucial to develop sound methodologies to conclusively ascertain the effectiveness of privileged information (PI). However, existing methods, being generic, are often mistakenly considered universal solutions. This misconception leads to a lack of thorough theoretical and empirical foundations regarding the impact of privileged information.

The key intuition behind LUPI is that privileged information should be addressed via *knowledge transfer* – transferring knowledge from the space of privileged information (**PI** model) to the space where the decision rule is constructed (**no-PI** model) (Vapnik & Izmailov, 2015b). State-of-the-art approaches for LUPI are largely based on two knowledge transfer techniques: *knowledge distillation* (Lopez-Paz et al., 2016; Markov & Matsui, 2016; Lee et al., 2020; Xu et al., 2020; Yang et al., 2022b) and *marginalization with weight sharing* (Lambert et al., 2018; Collier et al., 2022; Ortiz-Jimenez et al., 2023). In this work, we analyze these two popular knowledge transfer techniques for LUPI from both theoretical and practical perspectives.

Recent research suggests that incorporating PI is crucial for enhancing sample efficiency and generalization performance (Lambert et al., 2018; Yang et al., 2022b; Collier et al., 2022). These studies attempt to explain under what conditions LUPI is beneficial. However, theoretical analyses often either assume knowledge transfer occurs or demonstrate it takes place for extreme cases under assumptions that are difficult to verify. Additionally, empirical analyses in existing studies frequently rely on stylized examples (Collier et al., 2022; Ortiz-Jimenez et al., 2023), specific experimental settings (Lopez-Paz et al., 2016; Xu et al., 2020; Collier et al., 2022), or low-data regimes (Vapnik & Izmailov, 2015b; Markov & Matsui, 2016; Lopez-Paz et al., 2016; Lambert et al., 2018). Therefore, conclusively identifying that knowledge transfer happens and is induced by PI is non-trivial, and there remains a gap in understanding PI.

In this paper, we investigate whether knowledge transfer truly takes place in *knowledge distillation* and *marginalization with weight sharing*. To that end, we critically review the theory behind knowledge transfer in LUPI and explicitly discuss assumptions imposed by the existing theoretical analyses. We argue that the imposed assumptions are overly restrictive and discover that discussions on the robustness of the results to violations of these assumptions are frequently omitted. On the empirical side, we conduct an elaborate ablation study and demonstrate the apparent improvements often result from factors unrelated to PI. We reveal that previous studies tend to misinterpret the observed gains in empirical performance and mistakenly attribute them to PI. Interestingly, when focusing on the mechanisms that disclose **PI** models' better performance, we observe that the gap between **PI** and **no-PI** models can be bridged by simply training models longer or replacing PI with a constant.

Back to the real world, we validate the existing methods on four real-life datasets from various application domains, including e-commerce, healthcare, and aeronautics. Our results demonstrate that the state-of-the-art approaches fail to outperform a model that does not use PI, which adds evidence to the limited contributions of LUPI in practical applications. Overall, our study highlights that, in the current state of research, there is no solid empirical or theoretical evidence that knowledge transfer takes place in the LUPI paradigm.

**Our contribution**  Our key contributions can be summarized as follows:

- We critically review the theory behind knowledge transfer in LUPI and argue that current research provides little theoretical justification for when LUPI should work.

---

performance indicator for understanding and optimizing customer journeys. This metric is widely utilized in practice Farris et al. (2016); Purnomo (2023).

- We revisit empirical studies that claim performance improvements due to PI and highlight that these improvements can be explained through mechanisms unrelated to PI.

- We conduct experiments on four real-world datasets from various application domains and find out that *no* improvement from **PI** model is observed, which adds evidence to the limited contribution of LUPI in practical applications.

Concerns about LUPI are not unprecedented. Earlier work by Serra-Toro et al. (2014) discusses experiments on SVM+, one of the first algorithms developed for LUPI (Vapnik & Vashist, 2009), that yield identical results to the regular Support Vector Machine (SVM) algorithm with randomly generated features as PI. Our analysis extends to newer algorithms that utilize PI, further advancing our understanding of the practical limitations of LUPI algorithms despite recent developments.

**Paper outline**   The rest of the paper is organized as follows. Section 2 discusses the knowledge transfer in LUPI and introduces the techniques of knowledge distillation and marginalization with weight sharing. Section 3 reviews the theory behind knowledge transfer in LUPI. The common misinterpretations are outlined in Section 4, with elaborate analyses of knowledge distillation in Section 4.1 and marginalization with weight sharing in Section 4.2. This is followed by our real-world experiments in Section 5. Finally, we conclude with Section 6. The source code of the experiments can be found at `https://github.com/danilprov/rethinking_lupi`.

## 2   Knowledge transfer in LUPI

In this section, we present two popular knowledge transfer techniques that are largely used in LUPI.

Let $\mathcal{D}$ denote a training dataset, $\mathcal{D} := \{(x_i, z_i, y_i)\}_{i=1}^{n}$, consisting of triples: features $x_i \in \mathcal{X}$, available during both training and inference, privileged information $z_i \in \mathcal{Z}$ accessible only during training, and labels $y_i \in \mathcal{Y}$ drawn from the unknown distribution $p(\cdot|x_i, z_i)$. We focus on a $c$-class classification task (i.e., $y_i \in \{1, \ldots, c\}$), although the same ideas apply to a regression task.

The LUPI problem is often described as an interaction between an intelligent teacher, who has access to PI, and a student, who learns from the teacher's 'explanations' (Vapnik & Vashist, 2009). Let $\mathcal{G}_t := \{g | g : \mathcal{X} \times \mathcal{Z} \to \mathcal{Y}\}$ be a teacher function class and $\mathcal{G}_s := \{g | g : \mathcal{X} \to \mathcal{Y}\}$ be a student function class. Vapnik & Izmailov (2015b) formulate two conditions that are required to learn effectively using PI:

1. the empirical error in privileged space $\mathcal{X} \times \mathcal{Z}$ is smaller than the empirical error in the feature space $\mathcal{X}$, i.e., the classification rule $y = g_t(x, z)$ is more accurate than the classification rule $y = g_s(x)$, for some $g_t \in \mathcal{G}_t$ and the best $g_s \in \mathcal{G}_s$.

2. the knowledge of the rule $y = g_t(x, z)$ in space $\mathcal{X} \times \mathcal{Z}$ can be represented/transferred to improve the accuracy of the desired rule $y = g_s(x)$ in space $\mathcal{X}$.

Assuming that the first condition holds, which is easy to verify empirically on a given dataset, the difficulty is to verify whether and when the knowledge transfer actually happens. To address this challenge, two main knowledge transfer techniques have been proposed in the LUPI literature for improving the accuracy of rule $y = g_s(x)$: *knowledge distillation* and *marginalization with weight sharing.*

**Knowledge distillation**   Distillation introduced by Hinton et al. (2015) forms the basis for knowledge distillation methods using PI (Lopez-Paz et al., 2016; Markov & Matsui, 2016; Garcia et al., 2020; Lee et al., 2020; Xu et al., 2020; Yang et al., 2022b). Lopez-Paz et al. (2016) unifies LUPI with distillation for supervised learning and suggests that the representation learned by the **PI** model can be effectively distilled to a **no-PI** model. Their method, called Generalized distillation, proceeds in two stages. First, train a teacher model that takes both $x$ and $z$ as input to predict $y$. With a slight abuse of notation, we assume that $y$ is represented by a one-hot encoded vector, i.e., $y \in \Delta^c$, where $\Delta^c$ is a set of $c$-dimensional probability vectors. The teacher's

goal is to learn the representation

$$g_t = \underset{g \in \mathcal{G}_t}{\arg\min} \frac{1}{n} \sum_{i=1}^{n} \ell\left(y_i, \sigma(g(x_i, z_i))\right), \tag{1}$$

where $\ell : \Delta^c \times \Delta^c \to \mathbb{R}_+$ is a loss function, and $\sigma : \mathbb{R}^c \to \Delta^c$ is the softmax operation.

In the second stage, a student model *distills* the learned representation $g_t$ into

$$g_s = \underset{g \in \mathcal{G}_s}{\arg\min} \frac{1}{n} \sum_{i=1}^{n} \left[(1 - \lambda)\ell\left(y_i, \sigma(g(x_i))\right) + \lambda\ell\left(s_i, \sigma(g(x_i))\right)\right], \tag{2}$$

where $s_i = \sigma(g_t(x_i, z_i)/T) \in \Delta^c$ is a soft label with temperature $T$ provided by the teacher model and $\lambda \in [0, 1]$ is the imitation parameter, which balances the importance between imitating the soft predictions $s_i$ and predicting the true hard labels $y_i$.

Intuitively, the teacher reveals the label dependencies to the privileged information by softening the class-probability predictions in $s_i$, and the student distills this knowledge by training using the input-output pairs $\{(x_i, y_i)\}_{i=1}^{n}, \{(x_i, s_i)\}_{i=1}^{n}$. The soft labels $s_i$ provided by the teacher assumed to contain more information than hard labels $y_i$ and allow faster learning (Lopez-Paz et al., 2016). After distilling the privileged information, we can use the student model $g_s \in \mathcal{G}_s$ for prediction at test time.

Generalized distillation underpins numerous PI algorithms (Markov & Matsui, 2016; Garcia et al., 2020; Lee et al., 2020; Xu et al., 2020; Yang et al., 2022b) introduced with problem-specific adjustments peripheral to the knowledge distillation component.

**Marginalization and weight sharing** Another popular approach of incorporating privileged information is based on marginal distribution $p(y|x) = \int p(y|x, z)p(z|x)dz$ (Lambert et al., 2018; Collier et al., 2022; Ortiz-Jimenez et al., 2023). Consider a training problem:

$$g_t = \underset{g \in \mathcal{G}_t}{\arg\min} \frac{1}{n} \sum_{i=1}^{n} \ell\left(y_i, g(x_i, z_i)\right). \tag{3}$$

This is equivalent to a classical supervised learning problem defined over the privileged space $\mathcal{X} \times \mathcal{Z}$. In order to solve the inference problem, we can consider the following marginal distribution

$$g_s(x) = \mathbb{E}_{z \sim p(z|x)} \left[g_t(x, z)\right]. \tag{4}$$

The major problem in this formulation is the intractability of computing the expectation in equation 4, as $p(z|x)$ is unknown. As such, Collier et al. (2022) propose a knowledge transfer technique based on weight sharing to approximate equation 4. Their method, called TRAM (transfer and marginalize), is designed to reduce the harmful impact of noisy labels and facilitate learning. The authors motivate their work by the ability of PI to reduce the effect of malicious or lazy annotators on collected labels.

TRAM is based on a two-headed model in which one head has access to PI, and the other one does not. Specifically, they propose a neural network architecture which consists of three parts: shared feature extractor $\phi(x)$, No PI head $g_s(x')$, and PI head $g_t(x', z)$, where $\phi : \mathcal{X} \to \mathcal{X}'$ learns representation $x'$ of features $x$ for some representation space $\mathcal{X}'$. Then, they consider the following two-step approach:

$$\phi^*, g_t = \underset{g \in \mathcal{G}_t, \phi}{\arg\min} \frac{1}{n} \sum_{i=1}^{n} \ell\left(y_i, g(\phi(x_i), z_i)\right), \tag{5}$$

$$g_s = \underset{g \in \mathcal{G}_s}{\arg\min} \frac{1}{n} \sum_{i=1}^{n} \ell\left(y_i, g(\phi^*(x_i))\right). \tag{6}$$

Crucially, feature extractor $\phi^*$ is learned in equation 5 with access to PI. This weight sharing assumed to enable knowledge transfer to the network trained without PI in equation 6. At test time, only the No PI head is used for prediction.

## 3   When is knowledge transfer in LUPI proven theoretically?

In this section, we review the existing theoretical analyses of the LUPI paradigm. Recent work attempts to explain when LUPI is beneficial, but finding conclusive theoretical evidence for knowledge transfer using PI remains challenging. These theoretical analyses often depend on strong assumptions and lack discussion on when these are satisfied or violated.

LUPI was introduced as a technique that can leverage PI to distinguish between easy and hard examples, a concept closely tied to SVMs, where the difficulty of an example can be quantified by the slack variable (Vapnik & Vashist, 2009). For the case of SVMs, Vapnik & Izmailov (2015b) show that utilizing slack variables as privileged information can result in a generalization error bound with rate $O(\frac{1}{n})$ instead of $O(\frac{1}{\sqrt{n}})$. The motivation behind this is that SVM classification becomes separable after we correct for the slack values, which measure the degree of misclassification of training data points. [2] Since it is unlikely that the teacher is able to provide true slack variables, the idea of the SVM+ algorithm is to estimate slack variables and represent them by the teacher's decision rule $g_t$. Technically, the improved convergence rate holds under two conditions: (i) function class $\mathcal{G}_t$ has a smaller capacity than student's function class $\mathcal{G}_s$ and (ii) teachers' explanations $p(z|x)$ engender a convergence that is faster than $O(\frac{1}{\sqrt{n}})$. However, the sets of functions satisfying these conditions are confined to Reproducing Kernel Hilbert Space (RKHS) (Vapnik & Izmailov, 2015b), and their theoretical justifications does not generalize beyond SVMs with decision rules defined in RKHS.

On the last point, Lopez-Paz et al. (2016) argue that in Generalized distillation, the rate at which the student learns from the teacher's soft labels is faster than $O(\frac{1}{\sqrt{n}})$, since soft labels contain more information than hard labels per example, and should allow for faster learning. This requirement on the learning rate is rather strong and hard to satisfy in a general setting.

Generalized distillation was also analyzed in the semi-supervised learning setting by Yang et al. (2022b). The authors consider a problem where two datasets are available: $\mathcal{D}_{label} := \{(x_i, z_i, y_i)\}_{i=1}^n$ and $\mathcal{D}_{unlabel} := \{(x_i, z_i)\}_{i=1}^m$. Their distillation algorithm trains the teacher model using labeled dataset $\mathcal{D}_{label}$, which provides pseudo-labels for both the labeled and unlabeled datasets, $\mathcal{D}_{label}$ and $\mathcal{D}_{unlabel}$, respectively. Then, the student model is trained on the combined dataset $\mathcal{D}_{label} \cup \mathcal{D}_{unlabel}$ using the imputed pseudo-labels as targets. They theoretically demonstrate that their algorithm reduces estimation variance in the case of linear models with independent regular and privileged features and report improved empirical performance. However, the improvement appears to largely come from the semi-supervised aspect rather than PI-induced knowledge transfer. In Appendix A, we show that when we have no unlabelled data, the estimation variance of distillation actually slightly increases.

For the marginalization approach, Lambert et al. (2018) demonstrate that the convergence rate can be increased to $O(\frac{1}{n})$ for convolutional neural networks under a strict assumption that the variance of the model can be upper-bounded by $\delta$ for an arbitrarily small value of $\delta > 0$. The authors leave verifying this assumption as an open problem.

Meanwhile, Collier et al. (2022) formulate two conditions under which marginalization can achieve a lower empirical risk for a linear regression $y = \mathbf{x}^\top \mathbf{w} + \mathbf{z}^\top \mathbf{v} + \epsilon$, where $\mathbf{z} \sim p(\mathbf{z}|\mathbf{x})$ (i) the regression coefficients $\mathbf{v}$ have a large variance when explained only by the features $\mathbf{x}$ and (ii) privileged features $\mathbf{z}$ have a significant average component outside of the subspace spanned by the features $\mathbf{x}$. However, their analysis is intractable beyond this simple case, and hence, we cannot quantify such conditions in the general setting.

Overall, the existing theory either assumes that knowledge transfer occurs or identifies conditions under which it might happen in stylized linear models. However, conclusive theoretical evidence supporting knowledge transfer through PI remains lacking.

---

[2] Separable classification corresponds to a situation when there exists a function that separates the training data without errors and admits generalization error bound of the rate $O(\frac{1}{n})$. Conversely, in non-separable classification, there is no function that can separate training data without errors, and the generalization error bound is of the order $O(\frac{1}{\sqrt{n}})$ (Vapnik, 1998).

Table 1: Expanding the training size of Experiment 1 (Clean Labels) from Lopez-Paz et al. (2016). The effect of Generalized distillation wears off when the training size surpasses 1000 samples.

| Training size | Privileged | Generalized distillation | no-PI |
|---|---|---|---|
| 200 | 0.95 ±0.01 | 0.95 ±0.01 | 0.87 ±0.02 |
| 500 | 0.95 ±0.01 | 0.95 ±0.01 | 0.92 ±0.01 |
| 1000 | 0.95 ±0.01 | 0.95 ±0.01 | 0.94 ±0.01 |
| 2000 | 0.95 ±0.01 | 0.95 ±0.01 | 0.95 ±0.01 |

## 4 What does existing empirical evidence show?

In this section, we revisit the original experiments conducted with the introduction of Generalised distillation and TRAM. Our goal is to challenge PI-induced knowledge transfer for these two methods in the original experiments. In Section 4.1, we revisit four experiments from Lopez-Paz et al. (2016) (one of the experiments is deferred to Appendix B) to highlight potential limitations and misinterpretations from the aforementioned work. In Section 4.2, we revisit the experiments by Collier et al. (2022) to demonstrate that TRAM fails to explain the annotators' noise, and the observed improvements in empirical risk can be explained by the architecture of TRAM.

### 4.1 Generalized distillation

**Synthetic experiments from Lopez-Paz et al. (2016)** Lopez-Paz et al. (2016) ran four experiments to demonstrate the ability of Generalized distillation to transfer knowledge. These are simulations of logistic regression models repeated over 100 random partitions. For the two experiments that see positive effects of using Generalised distillation, the triplets $(x_i, z_i, y_i)$ are sampled from one of two generating processes:

| Experiment 1: Clean labels as PI | Experiment 3: Relevant features as PI |
|---|---|
| $x_i \sim \mathcal{N}(0, I_d)$ | $x_i \sim \mathcal{N}(0, I_d)$ |
| $z_i \leftarrow \langle \alpha, x_i \rangle$ | $z_i \leftarrow x_{i,J}$ |
| $\epsilon_i \sim \mathcal{N}(0, 1)$ | |
| $y_i \leftarrow \mathbb{I}\{(z_i + \epsilon_i) > 0\}$ | $y_i \leftarrow \mathbb{I}\{\langle \alpha, z_i \rangle > 0\}$ |

where $d$ is dimensionality of regular features, $d = 50$, $\alpha \in \mathbb{R}^d$ is the separating hyperplane, and set $J$, $J = 3$, is a subset of the variable indices $\{1, \ldots, d\}$ chosen at random but common for all samples. Both Generalized distillation and **no-PI** models are trained on 200 samples ($n = 200$), and the authors report a substantial improvement in accuracy (88% vs. 95% for Clean labels as PI and 89% vs. 97% for Relevant features as PI) testing models on 10000 test samples.

The results in Table 1 demonstrate that knowledge transfer via PI can enhance performance in low-data regimes, serving as a valuable proof-of-concept. However, this effect rapidly diminishes as the sample size increases relative to the dimensionality of $x$. For detailed results for Experiment 3, see Table 3.

It is also important to note that in both experiments, PI contains (almost) perfect information about the distance of each sample to the decision boundary, and from a practical perspective, obtaining such high-quality PI is improbable. In Experiment 1, PI encodes the exact distance, while in Experiment 3, PI encodes the relevant features used to calculate the distance (both cases align with the perfect knowledge of the slack variables in Vapnik & Izmailov (2015a)).

**MNIST experiment from Lopez-Paz et al. (2016)** The authors further demonstrate PI-induced knowledge transfer using an experiment with the MNIST dataset. In this experiment, the teacher learns from full 28x28 images while the student learns from downscaled 7x7 images. They conduct two experiments with 300 and 500 training samples, reporting significant improvement in classification accuracy compared to a model without PI. When revisiting these experiments, we found that the original experiment limited training

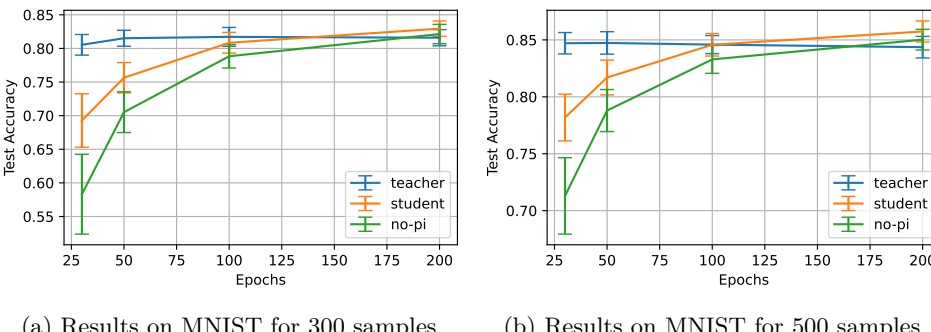

(a) Results on MNIST for 300 samples      (b) Results on MNIST for 500 samples

Figure 1: The effect of sufficient training epochs on the MNIST Generalised distillation experiment.

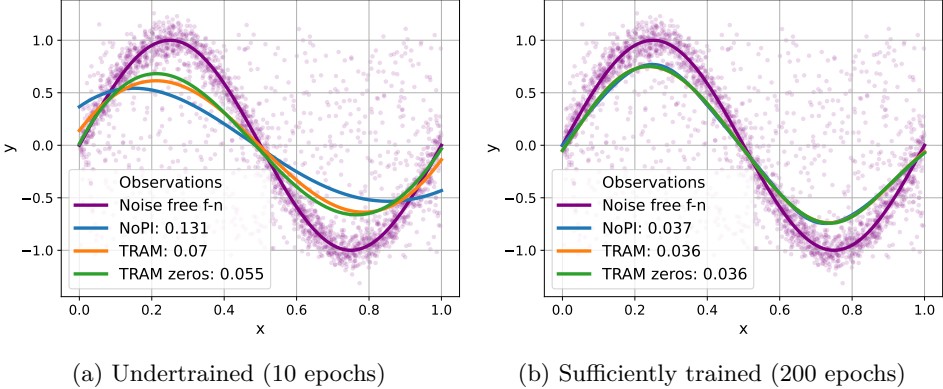

(a) Undertrained (10 epochs)      (b) Sufficiently trained (200 epochs)

Figure 2: TRAM zeros, TRAM, and **no-PI** for insufficient training (2a) and sufficient training (2b). The numbers in the legend indicate MSE loss with respect to the noise-free function.

epochs to 50. In Figure 1, we show that the reported effects are indeed visible around 50 epochs but quickly disappear when we allow all models to continue training.

Important to note, is that given the teacher-student setup, when the **no-PI** and student model performances are reported at 50 epochs in Figure 1, the student model actually requires a teacher model that had already completed 50 epochs, thus combined requiring 100 training epochs. Taking this into consideration, there is no evidence of either improved sample efficiency or computational efficiency by using Generalised distillation in this setting.

**Further discussion on knowledge distillation using PI**    While Generalized distillation shows preliminary evidence of knowledge transfer, we can see that it takes place only for low data regimes and in highly styled examples. To address these gaps, several attempts have been made from the application side (Markov & Matsui, 2016; Garcia et al., 2020; Lee et al., 2020; Xu et al., 2020), with Xu et al. (2020) applying generalized distillation to recommendations with privileged information in e-commerce. Admittedly, all of these works report marginal improvement over the **no-PI** model.

## 4.2 Revisiting TRAM

Collier et al. (2022) and Ortiz-Jimenez et al. (2023) argue that PI can be used to "explain away" label noise. To demonstrate TRAM having this capability, Collier et al. (2022) consider the following synthetic experiment: A noisy annotator $z$ is simulated by binary indicator $z \sim Ber(0.3)$, such that $z = 1$ represents the case where the noisy annotator provides a random label independent of $x$

$$y = (1 - z) \cdot \sin(2\pi x) + z \cdot v + \epsilon, \tag{7}$$

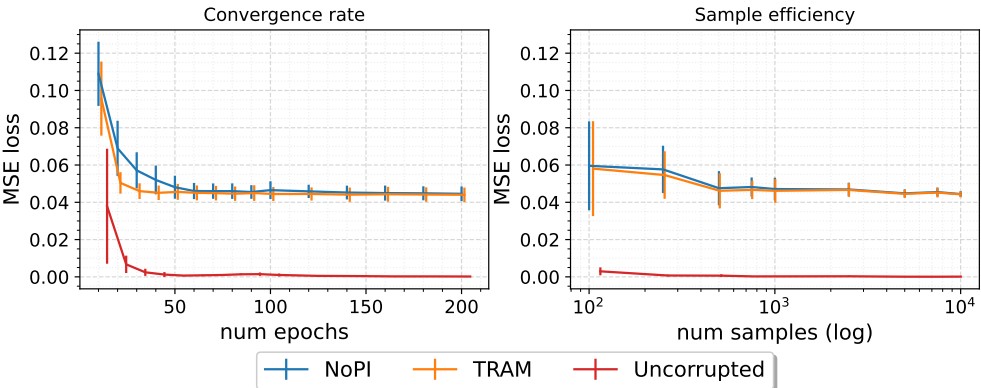

Figure 3: TRAM and **no-PI** training dynamics for the synthetic experiment from equation 7. **(Left)** presents training dynamics over 200 epochs. **(Right)** shows the resulting models' performances across varying sample sizes trained for 200 epochs. "Uncorrupted" corresponds to a regular model fitted to uncorrupted data $y = \sin(2\pi x) + \epsilon$.

where $x \in [0, 1], v \sim Unif(-1, 1)$, and $\epsilon \sim \mathcal{N}(0, 0.1)$.

The authors train TRAM and **no-PI** models on $n = 2500$ training samples using a 2-layer fully connected neural network with a tanh activation function. They observe results from Figure 2a and state "We see that the representations learned by the model with access to PI in step #1 [3] enable a near perfect fit to the true expected marginal distribution, $\mathbb{E}_{(z,y)\sim p(z,y|x)}[y]$, over $\mathcal{X}$. However, without access to PI, the noise term $a \cdot v$ cannot be explained away."

We regard the expression "explaining away the noise term" as cumbersome in this context: as one can see, neither TRAM nor **no-PI** effectively explains the noise term $z \cdot v$ away. The task of explaining noise term would ideally correspond to learning the noise-free function $\mathbb{E}[y|x, z = 0] = \sin(2\pi x)$; however, as depicted in Figure 2b, after sufficient training, TRAM and **no-PI** converge to a biased function. This effect is more clearly visible in Figure 3, where we compare the TRAM performance to an uncorrupted model (a regular model that is fitted to data without the corrupted labels coming from $v$). Thus, we can conclude that TRAM does not "average out" or "explain away" label noise; rather, similarly to the **no-PI** model, it completes the average $\mathbb{E}[y|x]$.

Next, we consider the training dynamics of TRAM against **no-PI** model for the regression task in equation 7. Figure 3 (left) shows the training dynamics over 200 epochs for $n = 2500$. Figure 3 (right) shows the models' performances trained for 200 epochs across varying numbers of samples. The $y$-axis represents the MSE loss with respect to the noise-free generating function $\sin(2\pi x)$.

Although, which was already observed in Figure 2b, both TRAM and **no-PI** models eventually converge to similar performance levels, some disparity is observed in their trajectories (refer to Figure 3 (left)), with TRAM achieving optimal performance generally faster (in Appendix D, we extend our analysis to classification tasks, which are generally more difficult, and the advantage of TRAM is more noticeable there). This suggests that TRAM has a faster convergence rate. However, from Figure 3 (right), we can see that both models enjoy the same performance after sufficient training, which suggests that TRAM is not more sample efficient than **no-PI** model. Thus, similar to the MNIST experiment, increasing the number of epochs for **no-PI** model achieves identical performance to TRAM, resulting in both models fitting the expected marginal distribution almost perfectly.

**Why TRAM does not leverage PI** In order to understand by which mechanisms TRAM enables a faster convergence rate, we consider a modification of TRAM, where instead of PI $z$, we plug in a zero vector (**TRAM zeros**). Figures 2a and 2b show that the performance of **TRAM zeros** is identical to the

---

[3]step #1 corresponds to learning feature extractor $\phi^*$ in equation 5.

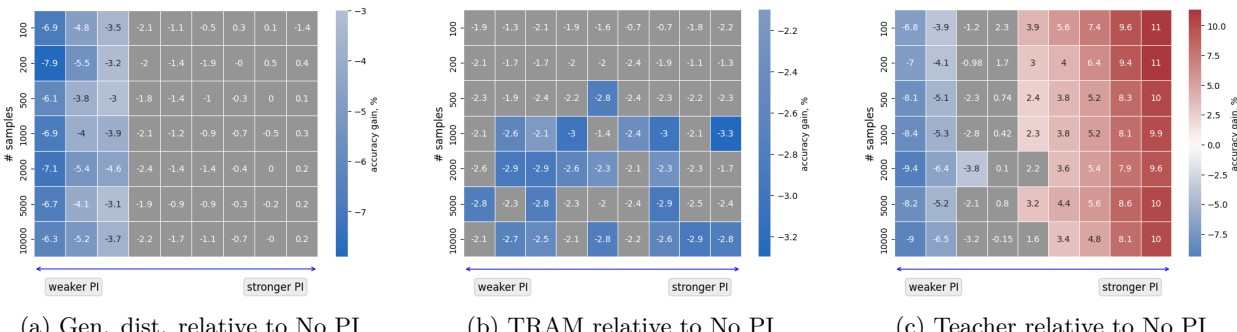

(a) Gen. dist. relative to No PI  (b) TRAM relative to No PI  (c) Teacher relative to No PI

Figure 4: Heatmap of accuracy gain (compared to No PI model; in %): warmer colors indicate positive gains, cooler colors indicate losses, grey cells correspond to statistically not significant results at 0.05. $x$-axis represents PI strength from stronger to weaker, and $y$-axis represents the number of samples. The results are averaged over 30 runs.

performance of TRAM using PI. This suggests that the benefit of TRAM stems from architectural changes rather than PI-induced knowledge transfer.

This mechanism can be traced back to the original TRAM experiments, as outlined in Appendix F of Collier et al. (2022). In this experiment, the authors reduced the capacity of the network by downsizing the number of parameters by 75% while keeping the number of training samples unchanged. Their observation indicated that TRAM performed equivalently to the **no-PI** model under these conditions. However, with the full-size network, TRAM exhibited a slight improvement over the **no-PI** model. This observation suggests that the full-size network might have been in an underfitted regime, where TRAM's architectural adjustments conferred an advantage.

### 4.3   Generalized distillation and TRAM for varying dataset sizes and PI strengths

The goal of this experiment is to evaluate the impact of PI on the overall performance across varying PI strengths and dataset sizes.

We generate a synthetic dataset with features normally distributed around the vertices of a $d$-dimensional hypercube and a binary label. In this experiment, $d = 20$ with 10 informative features and 10 redundant features. The regular features $x$ comprise 10 randomly selected features from the total 20. A subset of the features is treated as PI, which is unavailable to the No PI model. By varying the size of the PI features subset, we control the relative strength of PI compared to regular features. The weakest PI configuration includes one informative feature, while the strongest PI configuration includes all 10 informative features and the true label (!). Additionally, we vary the number of samples from 100 to 10,000.

For each combination of sample size and PI strength, we train each of the No PI, Gen. dist., TRAM and teacher models are used 30 times, and the relative performance is reported compared to the No PI model. Figure Figure 4a-4b presents heatmaps of the accuracy gain for the Gen. dist. and TRAM compared to the No PI model. We report the accuracy gain of the teacher model compared to the No PI model in Figure 4c as a sanity check to illustrate the potential of privileged information.

The results in Figure 4 highlight several key trends. For the teacher model (Figure 4c), weaker PI harms performance, moderate PI strength has negligible impact, and stronger PI leads to statistically significant improvements. For Gen. dist. and TRAM, the heatmaps reveal distinct patterns despite generally non-positive accuracy gains. Specifically, Gen. dist. (Figure 4a) performs worse in settings with weaker PI, whereas TRAM (Figure 4a) performs worse with larger sample sizes. Notably, neither Gen. dist. nor TRAM improves upon the No PI model, and both methods fail to demonstrate practical utility in this experimental setup.

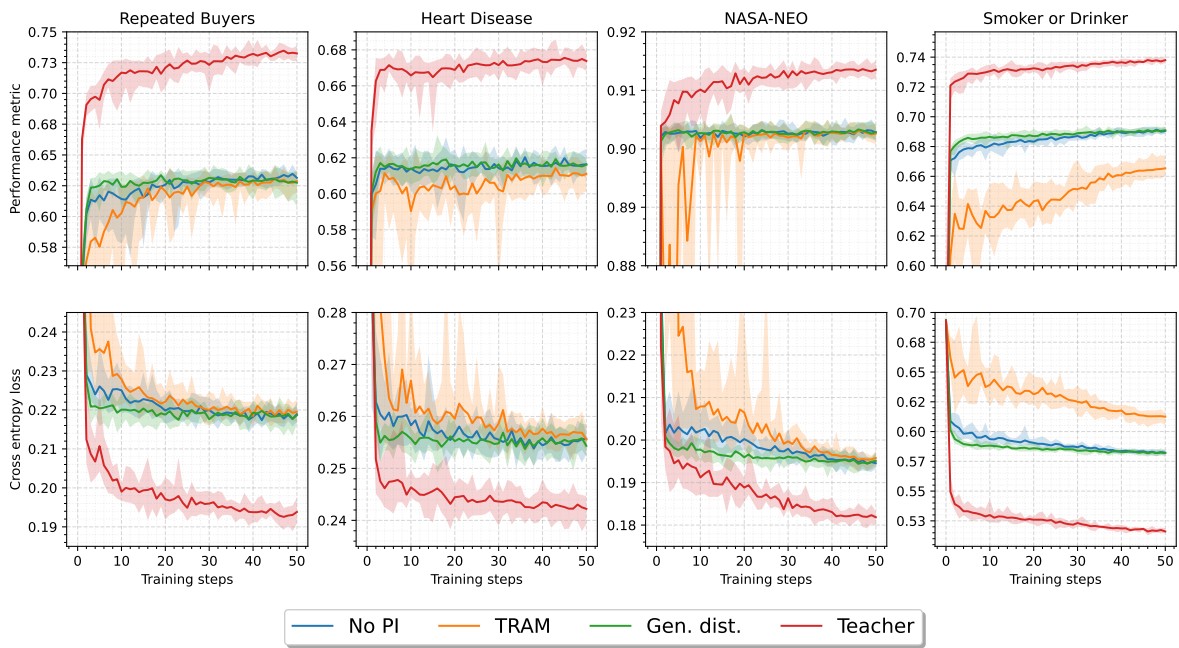

Figure 5: Training dynamics of No PI, TRAM, Gen. dist., and Teacher for 4 real-world datasets averaged over 10 runs. **(Top row)** shows the performance metric on the test set (scaled roc auc score for `Repeat Buyers` and `Heart Disease` datasets and accuracy for `NASA-NEO` and `Smoker or Drinker` datasets). **(Bottom row)** shows cross-entropy loss on the test set.

## 5  Real-world applications

To further validate the described methodologies, we conduct experiments on four real-world datasets from a variety of application domains, including e-commerce, healthcare, and aeronautics: [4]

- `Repeat Buyers` (Alibaba, 2024) Motivated by our use-case example, we consider the `Repeat Buyers` dataset, a large-scale public dataset from the IJCAI-15 competition. The data provides users' activity logs of an online retail platform, including user-related features, information about items at sale, and implicit multi-behavioral feedback such as *click*, *add to cart*, and *purchase*. We assign user-item features to $x$, intermediate signals *click* and *add to cart* to $z$, and *purchase* to $y$.

- `Heart Disease`  (BRFSS, 2024) This dataset is derived from the 2015 Behavioral Risk Factor Surveillance System, and it contains $\sim 260k$ cleaned responses, focusing on the binary classification of heart disease. We use social-demographic features (such as age and income) as privileged information $z$ and medical data as regular features $x$.

- `NASA-NEO` (NASA, 2024) NASA nearthest earth object dataset compiles the list of NASA-certified asteroids. It contains $\sim 90k$ samples with various properties of asteroids, and the task is to predict if an asteroid is hazardous. For the purpose of our study, we treat a subset of original features as privileged information.

- `Smoker or Drinker`  (Soo, 2024) This dataset was collected from the National Health Insurance Service in Korea. It compiles medical histories of $\sim 900k$ patients, focusing on their smoking and drinking status. For our study, we treat a subset of original features as privileged information.

The choice of PI in our experiments was motivated by three practical use cases of PI: (1) PI as features that materialize post-inference, (2) PI as features that are expensive to compute during real-time inference, and

---

[4]All datasets are distributed under CC BY-NC 4.0 license.

Table 2: Comparison of models' performance on test data. Results represent MEAN ± STD. DEV. and are averaged over 10 random seeds. We use scaled roc auc score for `Repeat Buyers` and `Heart Disease` datasets and accuracy for `NASA-NEO` and `Smoker or Drinker` datasets.

| Dataset | Method | ↓ Cross-entropy loss | ↑ Metric | |
|---------|--------|----------------------|----------|---|
| Repeat Buyers | **no-PI** | 0.2189 ± 0.0015 | 63.13 ± 0.25 | |
| | TRAM | 0.2194 ± 0.0013 | 62.89 ± 0.42 | |
| | Gen. dist. | 0.2183 ± 0.0017 | 62.88 ± 0.56 | sc. roc auc |
| | Teacher | 0.1938 ± 0.0019 | 73.23 ± 0.38 | |
| Heart Disease | **no-PI** | 0.2557 ± 0.0020 | 61.64 ± 0.47 | |
| | TRAM | 0.2555 ± 0.0018 | 61.62 ± 0.30 | |
| | Gen. dist. | 0.2543 ± 0.0013 | 61.11 ± 0.42 | |
| | Teacher | 0.2422 ± 0.0018 | 67.38 ± 0.31 | |
| NASA-NEO | **no-PI** | 0.1945 ± 0.0009 | 90.28 ± 0.07 | |
| | TRAM | 0.1948 ± 0.0009 | 90.26 ± 0.09 | |
| | Gen. dist. | 0.1951 ± 0.0010 | 90.29 ± 0.09 | accuracy |
| | Teacher | 0.1818 ± 0.0011 | 91.35 ± 0.10 | |
| Drinker or Smoker | **no-PI** | 0.5823 ± 0.0017 | 69.05 ± 0.15 | |
| | TRAM | 0.6125 ± 0.0031 | 66.54 ± 0.44 | |
| | Gen. dist. | 0.5820 ± 0.0009 | 69.09 ± 0.15 | |
| | Teacher | 0.5157 ± 0.0011 | 73.08 ± 0.11 | |

(3) PI as features that might be unavailable (e.g., due to privacy reasons). These use cases are prevalent in practice, and our choice of PI in experiments was specifically aligned with them. For instance, the Repeated Buyers dataset corresponded to the first use case, the Heart Disease, and NASA-NEO datasets to the second, and the Smoker or Drinker datasets to the third.

We consider Generalized distillation, TRAM, and **no-PI** models, which are 2-layer fully-connected neural networks for all datasets. For reference, we report the teacher's performance for all datasets to indicate that PI could be useful in all cases. We perform a timestamp-based train test split and use 70% of data for training each model and 30% of data for reporting performance. The experiments are repeated over 10 random model initialization. The further experimental details are provided in Appendix E.

Figure 5 shows the training dynamics for TRAM, Generalized distillation, and **no-PI** models across the four datasets, and Table 2 reports the resulting performance metric. We use scaled roc auc[5] for `Repeat Buyers` and `Heart Disease` datasets and accuracy for `NASA-NEO` and `Smoker or Drinker` datasets. As we can see, there is no benefit from using TRAM or Generalized distillation over **no-PI** model for all datasets, with TRAM performing substantially worse in `Smoker or Drinker` dataset. Therefore, there is no evidence that TRAM and Generalized distillation transfer knowledge from privileged information, and there is no added value in a real-world setting with moderate to large data sizes and properly tuned and trained models.

## 6  Conclusion

Misinterpretations of empirical results and attributing performance gains to privileged information are so prevalent in recent literature that they create a widely accepted impression of the uncompromising usefulness of PI. However, this misconception has resulted in a lack of thorough theoretical and empirical foundations regarding the true impact of privileged information.

Our theoretical overview of recent developments in LUPI argues that the existing theory does not provide a sufficient basis for claiming that knowledge transfer occurs and highlights the need for a more solid theoretical justification. While this observation only applies to the theoretical analyses discussed in our study, we are also unaware of other work compellingly showing when knowledge transfer is possible and effective in LUPI.

---

[5]scaled roc auc = 2 * roc auc - 1

Through our experiments, we identify common fallacies of misinterpreting gains in empirical performance as knowledge transfer induced by PI. We demonstrate that after adequate training, state-of-the-art LUPI methods fail to outperform **no-PI** model. Surprisingly, we observe that low data regimes and undertrained models (low training epoch regimes) often seem to be confused. While PI is beneficial in low data regimes in highly styled examples, it has yet to be verified that this can be extended to realistic settings. So far, existing methods benefit from other factors unrelated to PI.

Similarly, our empirical evidence for TRAM suggests a lack of support for the notion that PI accounts for noise originating from corrupted labels. We illustrate that the purported improvements in empirical risk achieved through TRAM can be attributed to alterations in model architecture.

Addressing these concerns, our empirical and theoretical analyses provide compelling evidence that existing methods are insufficient in achieving effective learning using PI in practical, realistic scenarios; thereby rethinking knowledge transfer in learning using privileged information.

Nevertheless, our findings do not definitively disprove the possibility of knowledge transfer induced by privileged information. While the prevailing view on knowledge transfer in LUPI methods centers on enhancing general model performance through metrics such as accuracy or loss, our findings invite a broader perspective. The anticipated performance gains from LUPI methods may not materialize in general settings; however, PI could still play a critical role in more specific domains. For example, by providing additional context, PI might be leveraged to improve fairness in machine learning models – helping mitigate biases in decision-making processes (Chai et al., 2022; Yan et al., 2023). This suggests that a sole focus on performance-based metrics may overlook other valuable benefits of LUPI, emphasizing the need for research that explores its potential.

Thus, we believe practitioners and researchers should exercise caution when working with PI to avoid potential performance degradation or unintended inductive biases caused by experiment setup or dataset anomalies. Additionally, we urge the research community to devise more sound methodologies to conclusively ascertain the presence and effectiveness of knowledge transfer induced by PI.

## Acknoledgements

Part of this work was carried out during DP's internship at Booking.com. This project is partially financed by the Dutch Research Council (NWO) and the ICAI initiative in collaboration with KPN. The authors thank Philip Boeken and Andrey Davydov for discussions on earlier drafts of the paper.

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

## A Independent features

We follow the proof by Yang et al. (2022b) for the special case that $m = 0$ (no unlabelled instances)

Assuming a linear model generating the label $y$ as follows:

$$y = \mathbf{x}^\mathsf{T}\mathbf{w}^* + \mathbf{z}^\mathsf{T}\mathbf{v}^* + \epsilon, \quad \epsilon \sim \mathcal{N}(0, \sigma^2), \tag{8}$$

where $\mathbf{w}^* \in \mathbb{R}_x^d$ and $\mathbf{v}^* \in \mathbb{R}_z^d$ the unknown parameters, the regular features $x \sim \mathcal{N}(0, I_{d_x})$, the privileged features $z \sim \mathcal{N}(0, I_{d_z})$. and $\epsilon$ represents label noise. The solution of the standard linear regression is

$$\hat{\mathbf{w}}_{\mathrm{reg}} = \mathbf{X}^\dagger y = \mathbf{X}^\dagger(\mathbf{X}\mathbf{w}^* + \mathbf{Z}\mathbf{v}^* + \mathbf{N}) = \mathbf{w}^* + \mathbf{X}^\dagger(\mathbf{Z}\mathbf{v}^* + \mathbf{N}), \tag{9}$$

where $N \in \mathbb{R}^{n \times 1}$ the label noise vector. Therefore, we have

$$\mathbb{E}_{\mathbf{X}}\|\hat{\mathbf{w}}_{\mathrm{reg}} - \mathbf{w}^*\|_2^2 = \mathbb{E}_{\mathbf{X}}\|(\mathbf{Z}\mathbf{v}^* + \mathbf{N})^\mathsf{T}\mathbf{X}^{\dagger\mathsf{T}}\mathbf{X}^\dagger(\mathbf{Z}\mathbf{v}^* + \mathbf{N})\|_2^2$$
$$= \frac{d_x \cdot (\sigma^2 + \|\mathbf{v}^*\|^2)}{n - d_x - 1}$$

The last equality holds because $\mathbf{X}^{\dagger\mathsf{T}}\mathbf{X}^\dagger = (\mathbf{X}^\mathsf{T}\mathbf{X})^{-1}$ follows the inverse-Wishart distribution, whose expectation is $\frac{I_{d_x}}{n - d_x - 1}$.

For generalised distillation, the teacher $\hat{\theta} \in \mathbb{R}^{d_x + d_z}$, we have

$$\hat{\theta} = [\mathbf{X}; \mathbf{Z}]^\dagger [\mathbf{X}\mathbf{w}^* + \mathbf{Z}\mathbf{v}^* + \mathbf{N})]$$
$$= [\mathbf{w}^{*\mathsf{T}}; \mathbf{w}^{*\mathsf{T}}]^\mathsf{T} + [(\mathbf{X}_{\mathbf{Z},\perp}\mathbf{N})^\mathsf{T}; (\mathbf{Z}_{\mathbf{X},\perp}\mathbf{N})^\mathsf{T}]^\mathsf{T},$$

where $\mathbf{X}_{\mathbf{Z},\perp}$ is the pseudo inverse of the projection of X to the column space orthogonal to $\mathbf{Z}$, and $\mathbf{Z}_{\mathbf{X},\perp}$ is defined similarly. After distillation, we have that

$$\hat{\mathbf{w}}_{\mathrm{pri}} = \mathbf{X}^\dagger [\mathbf{X}; \mathbf{Z}] \hat{\theta}$$
$$= \hat{\mathbf{w}}^* + \mathbf{X}^\dagger\mathbf{Z}\hat{\mathbf{v}}^* + \mathbf{X}_{\mathbf{Z},\perp}^\dagger\mathbf{N} + \mathbf{X}^\dagger\mathbf{Z}\mathbf{Z}_{\mathbf{X},\perp}^\dagger\mathbf{N}.$$

We note that $\mathbf{Z}_{\mathbf{X},\perp}^\dagger\mathbf{N}$ has variance of order $\mathcal{O}\left(\frac{1}{n^2}\right)$, which is a non dominating term. For the other two terms we have

$$\mathbb{E}_{\mathbf{X},\mathbf{Z}}\|\hat{\mathbf{w}}_{\mathrm{pri}} - \mathbf{w}^*\|_2^2 = \mathbb{E}_{\mathbf{X},\mathbf{Z}}\|\mathbf{X}^\dagger\mathbf{Z}\mathbf{v}^* + \mathbf{X}_{\mathbf{Z},\perp}^\dagger\mathbf{N}\|_2^2$$
$$= \frac{d_x \cdot \|\mathbf{v}^*\|^2}{n - d_x - 1} + \frac{d_x \cdot \sigma^2}{n - d_x - d_z - 1}$$
$$\geq \mathbb{E}_{\mathbf{X}}\|\hat{\mathbf{w}}_{\mathrm{reg}} - \mathbf{w}^*\|_2^2.$$

## B Generalized distillation: SARCOS experiment

**SARCOS experiment from Lopez-Paz et al. (2016)** The last experiment provided by Lopez-Paz et al. (2016) is based on the SARCOS dataset (Vijayakumar, 2000). This dataset characterizes the 7 joint torques

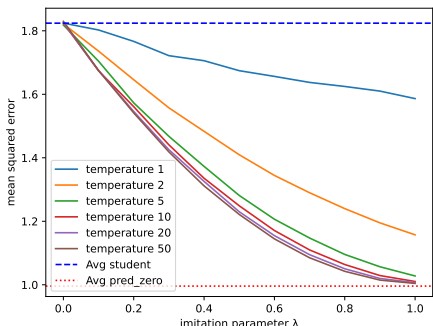

Figure 6: Reproducing the SARCOS experiment with the teacher replaced with $g_t = 0$.

of a robotic arm given 21 real-valued features. Lopez-Paz et al. (2016) learns a teacher on 300 samples to predict each of the 7 torques given the other 6, and then distills this knowledge into a student who uses as her regular input space the 21 real-valued features. They report improvement in mean squared error when using Generalized distillation and conclude, "when distilling at the proper temperature, distillation allowed the student to match her teacher performance."

However, there is a misalignment between the experiment setup and the conclusion drawn by the authors. It is observed that as the teacher labels approach 0, the student's performance improves. In fact, in Figure equation 6, we demonstrate that, due to the experiment setup, plugging in all zeros as a target for the student model corresponds to the best student's performance. In their code, instead of applying $T$ as a softmax temperature to the labels, the authors divide the soft label by $T$. This means that by increasing the temperature $T$ and the imitation parameter $\lambda$ in the original experiment, the authors force the teacher labels closer to 0 and report the observed improvement. Given that this is not reported in the paper, we believe this to be unintended by the authors. However, this means that the performance improvement can fully be attributed to the temperature scaling and not to a successful knowledge transfer of PI.

## C   Experiment 3

Table 3: Expanding the training size of Experiment 3 (Relevant features as privileged information) from Lopez-Paz et al. (2016). The effect of Generalized distillation wears off when the training size surpasses 2000 samples.

| Training size | Privileged | Generalized distillation | no-PI |
|---|---|---|---|
| 200 | 0.97 ±0.02 | 0.96 ±0.02 | 0.84 ±0.03 |
| 500 | 0.97 ±0.02 | 0.97 ±0.01 | 0.92 ±0.02 |
| 1000 | 0.98 ±0.02 | 0.97 ±0.01 | 0.95 ±0.01 |
| 2000 | 0.98 ±0.02 | 0.97 ±0.01 | 0.96 ±0.01 |
| 5000 | 0.98 ±0.02 | 0.97 ±0.01 | 0.97 ±0.01 |

## D   Extending the TRAM experiment to classification tasks

**Synthetic experiments for classification task**   To further demonstrate that explaining away harmful noise is non-trivial, extend the setting above to a classification task to make it more suitable for our use-case example. As such, $y$ is a binary label that represents conversion, and $z$ is PI, which represents the nature of the click.

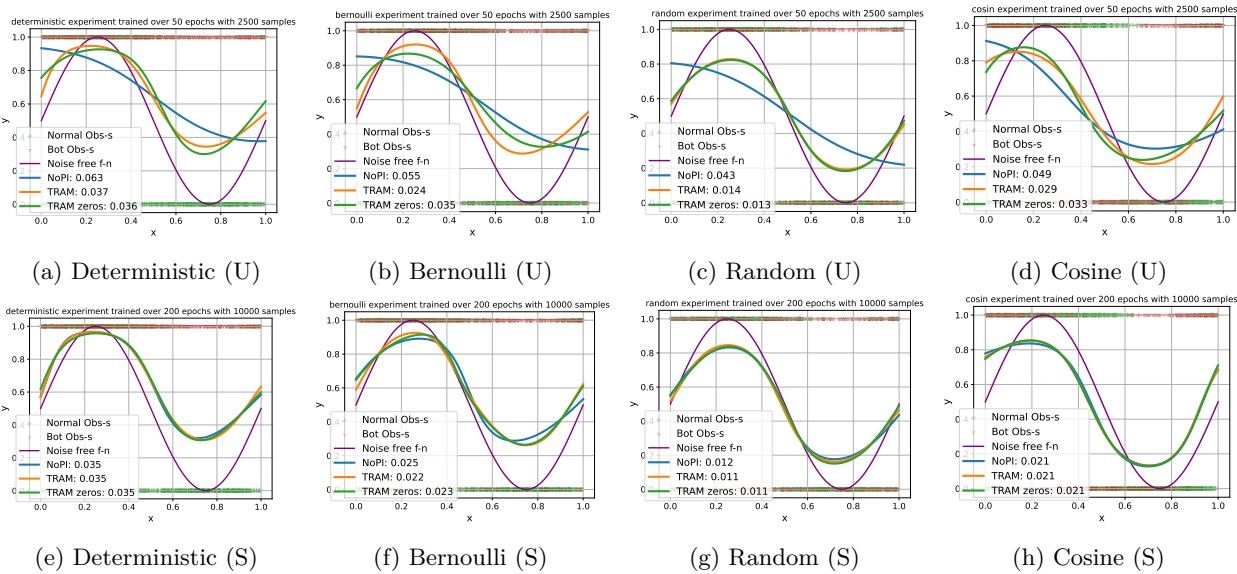

Figure 7: Example of TRAM and **no-PI** for 4 classification tasks. The models are trained for 50 epochs and 2500 samples in the top row and for 200 epochs and 10000 samples in the bottom row. The numbers in the legend indicate MSE loss with respect to the noise-free function. (U) corresponds to an undertrained regime, (S) corresponds to a sufficiently trained regime.

Similarly to Collier et al. (2022), $z \sim Ber(0.3)$, and the data generating process is as follows:

$$y_{score} = (1 - z) \cdot \sin(2\pi x) + z \cdot v, \tag{10}$$
$$y \sim Ber(y_{score}),$$

where $x \in [0, 1]$ and $v$ represents the nature of the click. We consider four scenarios of PI impact on the label: *Deterministic* − $v = 1$, *Bernoulli* − $v \sim Ber(0.7)$, *Uniform* − $v \sim Unif[-1, 1]$, *Cosine* − $v = \cos(2\pi x)$. The examples of these scenarios and trained TRAM and **no-PI** models are represented in Figure 7, with Figures 7a-7d representing models trained for 50 epochs with 2500 samples and Figures 7e-7h representing models trained for 200 epochs with 10000 samples.

Intuitively, *Uniform* resembles the original setup of Collier et al. (2022) but for the classification task. In our setting, it can be motivated by a bot or users that just randomly click on banners. *Deterministic* might correspond to an adversary that, for example, always clicks and never makes a purchase. Intuitively, explaining the noise for *Deterministic* regime should be more difficult than for *Uniform* regime because there is no randomness. *Bernoulli* regime is a middle point between *Uniform* and *Deterministic* regimes – there is still corruption but with some randomness. Finally, *Cosine* corresponds to a scenario when there are two types of users with different click behavior (according to sin for part of the population and to cos for the rest of the population).

Taking a closer look at Figures 7a-7d, we can see that TRAM enables a faster convergence rate. However, from Figures 7e-7h, it is apparent that both models No PI and TRAM eventually converged to the same functions, which do not correspond to the noise-free function $\sin(2\pi x)$.

Finally, we empirically analyze the sample efficiency of TRAM compared to the **no-PI** model. We train TRAM and No PI models for various values of $n$, from 100 to 10000. Both models are trained for 200 epochs for each generated dataset. Figure 8 (right) presents MSE loss across different values of $n$. We can see that both models converge to roughly the same value of all data regimes and all values of $n$, which suggests that TRAM doesn't enhance the sample efficiency.

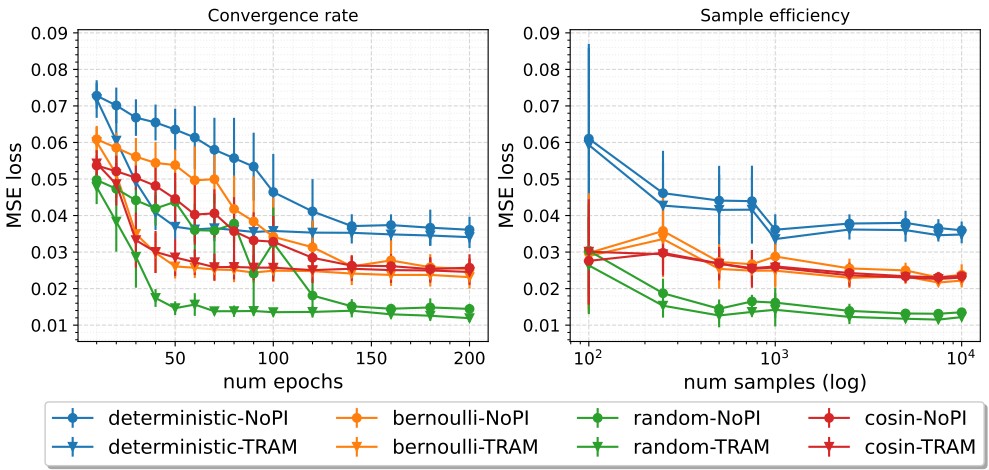

Figure 8: TRAM and **no-PI** training dynamics for 4 data regimes.

# E   Experimental details

This section describes experimental details for sections 4.1, 4.2, and 5. The source code for all experiments is attached in supplementary materials and will be available publicly upon acceptance of the article. We distribute all runs across 6 CPU nodes (Intel(R) CPU i7-10750H) and 1 GPU Nvidia Quadro T1000 per run for experiments.

**Generalized distillation experiments**   We follow the original setup of Lopez-Paz et al. (2016). For both Experiment 1 and Experiment 3, as a **no-PI**, student, and teacher models, we use 1 linear layer of dimension 50, with softmax activation. The networks were trained using an rmsprop optimizer with a mean squared error loss function. The temperature and imitation parameters for Generalized distillation were set to 1.

For MNIST and SARCOS experiments, we use two-layer fully connected neural networks of dimension 20, with ReLU hidden activations and softmax output activation for the **no-PI**, student, and teacher models. The MNIST experiment corresponds to a classification task with ten labels. The networks were trained using an rmsprop optimizer with a mean squared error loss function. The temperature and imitation parameters for Generalized distillation in the MNIST experiment were set to 10 and 1, respectively, as the best parameter set from the original paper Lopez-Paz et al. (2016).

**TRAM experiments**   We consider two tasks regression and classification. For both of them, as a **no-PI** model, we use two-layer fully connected neural networks of dimension 64, with tanh hidden activations and linear output activation for regression and sigmoid for classification. TRAM model has an extra hidden layer of size 64 with tanh activation function in the PI head. Both TRAM and **no-PI** networks are fit using the Adam optimizer Kingma & Ba (2017) with mean squared error loss function. The numbers of epochs are specified in figure captions for each experiment.

**Real-world experiments**   The experiment design is the same for all datasets unless stated otherwise.

For the no PI model, we use a two-layer fully connected neural network with the Gaussian error linear unit activation and a residual connection. For the Generalized distillation model, the teacher and student have the same architecture as the **no-PI** model, with teacher models' inputs $x$ and $z$ being fed independently to the linear layer first and then concatenated. The temperature and imitation parameters for Generalized distillation were set to 1 and 1, respectively, as the best parameter set. For TRAM, the feature extractor $\phi(x)$ also has an architecture of the **no-PI** model, and similarly to the teacher of Generalized distillation, the PI head of TRAM had independent inputs $x$ and $z$ that goes through a linear layer first.

All models are trained for 50 epochs with cross-entropy loss function and Adam optimizer with a base learning rate of 0.001, $\beta_1 = 0.9$, $\beta_1 = 0.95$, $\epsilon = 1e - 07$. All models are trained with L2 weight regularization with a decay weight of 0.1.

We train all models 10 times with the random initialization, and for all models, we report the cross-entropy loss value and performance metric on the test data – `Heart Disease` datasets and accuracy for `NASA-NEO` and `Smoker or Drinker` datasets and ROC AUC scaled between 0 and 1 (2*ROC AUC - 1) for `Repeat Buyers` (refer to Table 2). Additionally, we report the training dynamics of the cross-entropy loss value and performance metric on the test data in Figure 5. The teacher performance is provided for the reference to demonstrate that PI is indeed useful information.

## F    Other related work

**Multi-task learning**   While not strictly focused on the concept of PI, indications of successful knowledge transfer can be found in the field of multi-task (Caruana, 1997) and multi-objective learning (Mehrotra et al., 2020; Sagtani et al., 2024). The primary goal of this type of research is to find some joint- or Pareto optimal solution for multiple tasks or objectives simultaneously. These techniques could also be interpreted as a case of LUPI by predicting each privileged feature with an additional task. However, while instances of successful knowledge transfer have been reported in the literature, the quality of predictions is often observed to suffer with making multiple predictions due to a phenomenon called negative transfer (Standley et al., 2020).

Different from multi-task learning, LUPI mainly focuses on improving the learning of the target task rather than ensuring the performance of all the tasks (Jonschkowski et al., 2016). From the practical point of view, when using dozens of privileged features at once or when estimating the privileged features is more complicated than the original problem, it would be a challenge to tune all the tasks (Xu et al., 2020). For this reason, we focus on methods that can generalize to any type of PI and are not exclusive to auxiliary tasks.

**Surrogate signals**   In a similar spirit to LUPI, the proxy or surrogate signals literature (Athey et al., 2019; Mann et al., 2019; Yang et al., 2022a) studies how short-term outcomes can be used for estimating the long-term target outcome (e.g., in cancer studies). In this setting, the materialization of the target outcome is generally delayed to such an extent that it is unfeasible to use for decision-making. By using a short-term proxy or surrogate, existing work is able to construct a best-effort estimation of the primary signal before it has fully matured. In contrast to the PI setting, the issue of knowledge transfer is not presented. Additionally, we assume that the primary outcome has fully matured, hence the use of such proxies is not desirable.

