# OpenReview forum: "Rethinking Knowledge Transfer in Learning Using Privileged Information"
_TMLR — Accepted by TMLR_

### Review · Reviewer_wfft · 2024-11-30

**Summary Of Contributions:**

This manuscript examines the concept of Learning Using Privileged Information (LUPI), a machine learning paradigm where additional information is available during training but not during inference. The authors attempt to challenge the effectiveness of existing LUPI research from both theoretical and empirical approaches. Experimental validation across four real-world datasets from e-commerce, healthcare, and aeronautics domains, demonstrating that state-of-the-art LUPI methods fail to consistently outperform models without privileged information.

**Audience:**

Yes

**Broader Impact Concerns:**

There is no concern on the ethical implications of the work.

**Claims And Evidence:**

No

**Requested Changes:**

1. Expand the experimental design to include a broader range of LUPI techniques and more varied datasets and problem domains
2. Include more ablation studies to isolate the specific factors limiting LUPI effectiveness
3. Develop more sophisticated metrics for assessing knowledge transfer beyond traditional performance measures.

**Strengths And Weaknesses:**

Strengths:
1. Analysis spanning theoretical and empirical perspectives
2. Experiments across diverse real-world datasets

Weaknesses:
1. Limitations in Generalizability:
   - The theoretical analysis might not fully capture the nuanced ways privileged information could be beneficial
   - The conclusions might be overly broad given the limited set of methods examined
2. Potential Methodological Constraints and Confirmation Bias:
   - The experiments used relatively simple neural network architectures
   - The definition and selection of privileged information might influence results
   - The authors seem predisposed to proving the limitations of LUPI, which could lead to a selective interpretation of results
   - The experimental design might inadvertently be structured to minimize the potential benefits of privileged information
3. Potential Oversimplification
   - The conclusion that LUPI is ineffective might be premature
   - The study doesn't fully explore edge cases or specific domains where privileged information might be crucially important

---

### Review · Reviewer_DMQt · 2024-12-05

**Summary Of Contributions:**

Summary: The paper discusses learning with privileged information (PI) and raises concerns regarding the knowledge transform capabilities of such learning methods. The paper revisits two known methods: generalized distillation, and marginalization with weight sharing. The authors argue that the assumptions for theoretical analysis in these methods are over-restrictive, and empirically question the conclusions such works have made.

**Audience:**

Yes

**Broader Impact Concerns:**

No concern.

**Claims And Evidence:**

Yes

**Requested Changes:**

please see the weaknesses.

**Strengths And Weaknesses:**

**Strengths:**

S1. The paper is mostly well-written and structured.

S2. The paper discusses an important subject, learning with PI,  as it is being used in research frequently to improve generalization.

S3. The paper performs fairly sound evaluations, including experiments on real-world datasets.

**Weaknesses:**

W1. In the introduction section, par1, l 4, the authors say “… this information only materializes post-inference.” Can you elaborate on this and provide an example?

W2. In the same paragraph, “… optimizing a north-star metric such as product conversion”. Can you provide references?

W3. I believe the first contribution regarding the theoretical review of the LUPI methods is overstated. The paper simply re-states the (admitted) assumptions of the examined works and argues that they are strong or difficult to verify.

W4. In Section 4, in addition to the datasets, please provide more details on the underlying models, the task, and the learning algorithms.

W5. In Section 4.1, the presentation of experiments 1 and 2 could be improved. Embed them in a table for example.

W6. While the authors admit this in conclusion, it is worth repeating in Sections 1 and 4  that this paper only examines two LUPI methods in specific settings and refutes some of their claims.

W7. I wonder if the authors have any insight into the larger scales -- both larger datasets and larger models. Many works in deep learning use knowledge distillation and report improvements. Do you think larger feature space or parameter space would create the potential for knowledge transfer?

---

### Review · Reviewer_6Bsi · 2024-12-06

**Summary Of Contributions:**

The paper observes and critiques the justification of the LUPI paradigm from several earlier works.

**Audience:**

Yes

**Broader Impact Concerns:**

Not applicable.

**Claims And Evidence:**

Yes

**Requested Changes:**

I would not insist on essential changes right now. But in my view, the approach to analysing LUPI should be revised, involving the dependence of the contribution from PI on the training set size as an essential dimension.

**Strengths And Weaknesses:**

I believe that this critique of the LUPI paradigm is beneficial. It will prevent many users from wasting time because they trust in its universal applicability. The conceptual problem of PI is its non-asymptotic nature. As far as the training data set size grows, a machine-learning algorithm becomes able to work on its own, obtaining the same knowledge from non-privileged information.

On the other hand, for the same reason, I have to say a bit in defence of the LUPI paradigm. The first two rows of Table 1 (small training size) are valuable as a proof-of-concept. This is what transfer of PI can help with, that is, acceleration of learning at its early stages when the training set is small.

---

### Decision · Action_Editor_m9yA · 2025-02-07

**Recommendation:** Accept with minor revision

**Comment:**

Reviewers’ recommendations are a mix of acceptance and rejection.
We agree that the manuscript could be interesting and insightful for researchers working on privileged information; however, it still requires further refinement. The manuscript examines two Learning Using Privileged Information (LUPI) methods—generalized distillation and marginalization with weight sharing—in specific settings and refutes some of their claims. While Sections 1 and 4 have been revised to emphasize this scope based on the reviewers’ suggestions, the abstract still does not reflect this.

The conclusions might be overly broad given the limited set of methods examined. The concluding statement—“[o]ur work rethinks knowledge transfer in learning using PI and highlights that, in the current state of research, there is no solid empirical or theoretical evidence that LUPI works in realistic scenarios”—might be premature, as the study does not fully explore edge cases or specific domains where privileged information may be crucially important.

For example, Xu, Gao, Yu, and Darrell (“End-to-end Learning of Driving Models from Large-scale Video Datasets,” CVPR 2017) demonstrated how to “drive with privileged information.” In the discussion, the authors mentioned that privileged information could be leveraged to improve fairness in machine learning models; however, they did not include this discussion in the revised manuscript, as it does not “[enhance] general model performance relative to specific metrics such as accuracy or loss.” I encourage the authors to discuss this aspect further. For example, Yan, Odom, Pasunuri, Kersting, and Natarajan (“Learning with Privileged and Sensitive Information: A Gradient-Boosting Approach,” Frontiers in Artificial Intelligence, 2023) showed that their approach leverages sensitive information to improve performance while maintaining fairness.

**Audience:**

Yes.

**Claims And Evidence:**

The manuscript examines two learning using privileged information (LUPI) methods (generalized distillation and marginalization with weight sharing) in specific settings and refutes some of their claims.

---

> ### Author Response · Authors · 2025-02-26
> **Thank you to AE and Reviewers**
>
> We deeply appreciate the constructive feedback and insightful guidance provided by the action editor and reviewers. Thank you for helping strengthen our work and contributions.
>
> In the revised version, we have addressed all the concerns raised. Specifically:
> - We have refined the abstract to better reflect the scope of our study, explicitly emphasizing the two LUPI methods examined.
> - We have adjusted our conclusions to ensure they align with the specific methods studied and avoid overly broad claims.
> - We have expanded our discussion to acknowledge cases where privileged information may provide benefits beyond standard performance metrics.
>
> We have attached the camera-ready version of the paper.